# OpenReview forum: "Jigsaw Puzzles: Splitting Harmful Questions to Jailbreak Large Language Models in Multi-turn Interactions"
_colmweb.org/COLM/2025/Conference — COLM 2025_

### Official Review · Reviewer_KWaf · 2025-04-17

**Rating:** 3
**Confidence:** 4
**Ethics Flag:** 1

**Summary:**

This paper presents Jigsaw Puzzles (JSP), a multi-turn jailbreak strategy against large language models (LLMs). JSP splits harmful questions into harmless fractions and leverages multi-turn interactions to bypass defenses. Through experiments, the study highlights the vulnerabilities of LLMs in multi-turn settings and calls for the development of more robust defense mechanisms to enhance the safety and security of these models.

**Questions To Authors:**

See Reasons To Reject

**Reasons To Accept:**

- Well written and easy to understand approach.
- Multi-turn jailbreak attacks provide the community with greater warnings and insights, which can help developers of large models design more effective defense mechanisms.

**Reasons To Reject:**

- Lacking in innovation, decomposing harmful queries into seemingly innocuous ones for jailbreak attacks has already been proposed in previous work [1]. The motivation and methodology of this paper overlap significantly with those of the other work.
- It is inappropriate to claim that multi-turn condition jailbreak is under-explored [1-2].
- Insufficient discussion and experimentation: lacks discussion and experimentation with baselines [2].
- Prompts need to be carefully crafted, and I don't agree with such a high-cost approach. Is there a more flexible and efficient alternative you can provide?

[1] Chen, Zhaorun, et al. "Pandora: Detailed llm jailbreaking via collaborated phishing agents with decomposed reasoning." ICLR 2024 Workshop on Secure and Trustworthy Large Language Models. 2024.  \
[2] Li, Haoran, et al. "Multi-step jailbreaking privacy attacks on chatgpt." arXiv preprint arXiv:2304.05197 (2023).

---

> ### Author Response · Authors · 2025-05-30
>
> Thank you for your reviewing and suggestions.
>
> **Novelty:**
>
> Thank you for your sharing relevant works, we will include these into our related work section. Please read our response (**Comparison with other attacks**) to reviewer 9UUp. The process, **“decomposing harmful questions”**, in [1] and JSP is essentially different. [1] generates the corresponding benign sub-questions based on the original harmful question, which relies on a message chain to induce harmful responses step-by-step. While, JSP splits the original harmful questions into fractions as inputs of each turn. Besides this, the statement, “multi-turn condition jailbreak is under-explored”, is based on the comparison with single-turn jailbreak. We will clarify and update our expression in our paper.
>
> **Insufficient discussion and experiments:**
>
> Please read our response (**Comparison with other attacks** and **Diverse LLMs and Dataset Evaluation**) to reviewer 9UUp. We think the harmfulness of different jailbreaking strategies is case-by-case (i.e., fixing a SOTA attack method may be not helpful to mitigate other strategies’ risks, such as JSP). Therefore, in our work, we select other attack strategies (Red queen and Cipher) which have similarities with JSP to jointly reveal vulnerabilities and mechanisms (Section 4.5). Meanwhile we conduct our experiments on three commonly used harmful benchmarks (Section 4.1 and 4.4), demonstrating the consistent jailbreaking performance. And we also conduct experiments on the version updates of LLMs (Section 4.4) providing the insights on safety.
>
> **Jailbreak prompt:**
>
> Most current work on multi-turn jailbreaks (e.g. [1]) largely rely on the message chain, which needs a carefully designed prompt for each turn and users without professional knowledge can not easily abuse the vulnerabilities. However, compared with previous work on multi-turn strategies, JSP is already a more simple-to-implement method by laymans, the whole jailbreaking process can be realised via one LLM (attackers even can manually splitting harmful questions based on common sense), enabling an easier pathway to cause harm.

---

> > ### Comment · Reviewer_KWaf · 2025-06-06
> >
> > The author's response has not addressed my concerns, and this work still requires additional experiments or even further methodological innovations. I will maintain my score.

---

### Official Review · Reviewer_9UUp · 2025-05-11

**Rating:** 2
**Confidence:** 5
**Ethics Flag:** 1

**Summary:**

This paper introduces the Jigsaw Puzzles (JSP) strategy to exploit vulnerabilities in large language models (LLMs) through multi-turn interactions. Unlike most existing jailbreak methods that focus on single-turn prompts, JSP breaks harmful queries into harmless segments to evade content-based defenses. The JSP process involves three stages—query rewriting, sentence-level splitting, and word-level splitting—assisted by GPT-4. During the attack, a specially crafted prompt is followed by sequentially inputting the segments, with a final "Begin" command to elicit a response. Experimental results on five advanced LLMs demonstrate that JSP achieves high attack success rates and remains effective across various splitting strategies and defense mechanisms.

**Reasons To Accept:**

The research addresses a critical issue: while large language models (LLMs) are advancing rapidly, their security remains a fundamental concern. This paper focuses on jailbreak attacks in multi-turn interactions—a topic that has been largely underexplored in existing studies—making the research both timely and practically significant for enhancing LLM safety.

The experimental design is thorough and in-depth, covering five state-of-the-art LLMs and a variety of harmful questions, with evaluation based on metrics such as ASR-a and ASR-q. Through ablation studies, comparisons of interaction modes, and analysis of different splitting strategies, the paper provides a comprehensive assessment of the JSP strategy. These efforts offer rich empirical evidence and valuable insights for future research in LLM security.

**Reasons To Reject:**

1.Limited Generalizability of the Methodology
The paper evaluates the proposed JSP strategy on only five LLMs, three of which are developed by OpenAI. This narrow selection limits the generalizability of the results and fails to capture the diversity of the LLM landscape. I strongly recommend evaluating (1) more up-to-date models such as Gemini 2.5-Pro, and (2) models from other providers, such as Gemma3-27B and Qwen3-32B. Additionally, the study relies solely on the harmful question dataset introduced by Figstep. This dataset’s singularity may undermine the robustness and broader applicability of the experimental findings. It remains unclear how the strategy performs across models and threats of varying nature. (Questions: 1. Figstep is typically used in the MLLM domain—why was it selected over more established LLM safety datasets such as SafeBench or MM-SafetyBench? 2. For LLM safety evaluations, AdvBench and HarmBench are more commonly used; I recommend conducting experiments based on these benchmarks.)

2.Insufficient Coverage of Related Work
 Although the paper proposes a multi-turn jailbreak method, it does not discuss how it differs from other multi-turn jailbreak strategies, nor does it position itself within recent developments in the field. Key related works such as ActorAttack, CFA, MRJ, and RACE are not mentioned. Furthermore, the experimental section lacks a direct comparison with similar approaches, weakening the contextual relevance of the contribution.

3.Lack of Theoretical Depth
 The paper emphasizes the practical effectiveness of the JSP strategy but provides little theoretical insight into its underlying mechanisms. It does not adequately explain why the multi-turn fragmentation approach is able to bypass existing defenses, nor does it explore the fundamental vulnerabilities of LLMs under this attack paradigm. As a result, the theoretical foundation of the work is weak, and the core problem remains insufficiently examined.

4.Inadequate Discussion of Defense Mechanisms
While the paper demonstrates the effectiveness of the JSP attack against existing defenses, it does not offer any systematic or constructive proposals for improving these defenses. There is no exploration of how to adapt current defense strategies to account for the specific characteristics of JSP-style attacks. This lack of defensive insight undermines both the completeness and the practical utility of the research.

---

> ### Author Response · Authors · 2025-05-30
>
> Thank you for your reviewing and suggestions.
>
> **Diverse LLMs and Dataset Evaluation:**
>
> We would like to highlight our view that jailbreaking methods are not designed to bypass all models, as each model has its own strengths and weaknesses in terms of safety. In fact, any jailbreaking technique is likely to be effective only for a few weeks or months before being addressed by model developers. Therefore, this research, like any other work in this field, should be viewed as a necessary step toward building safer models that serve hundreds of millions of users. E.g., we conduct experiments on Claude models and o1 with specific safety modules, such as malicious intent detection and CoT (line 260-263), and JSP fails to jailbreak such models, we observe that such models can detect malicious intents from JSP prompt, which provide insights for the potential development of defence strategies. Meanwhile, we also have experiments on the version updates of LLMs (Section 4.4): e.g., the update from GPT-4o-0513 to GPT-4o-0806 exhibits potential effective defence mechanisms, while JSP reveals new vulnerabilities on Gemini-1.5-Pro-002 compared with Gemini-1.5-pro-001.
>
> Our work was inspired by Figstep (image splitting) and we directly conducted our strategy on it. Meanwhile, Figstep is a multimodal dataset, but it still provides enough harmful categories and diverse samples in text. From the perspective of datasets, we would like to clarify that we already conduct JSP experiments on Figstep, **Advbench**, and **Malicious-Instruct** (Section 4.1 and 4.4), covering diverse harmful categories and 339 harmful samples in total. Figstep, **Advbench**, and **Malicious-Instruct** are commonly used and standard harmful query benchmarks, and our results show consistent jailbreaking performance on these **three** dataset (Table 4), demonstrating JSP’s effectiveness and generalisability.
>
> **Comparison with other attacks:**
>
> Thank you for your sharing relevant works, we will include these into related work and discuss them deeply. However, we would like to emphasise that these works are essentially different from JSP (line 95), such multi-turn strategies rely on the message chain design and users without professional knowledge can not easily abuse the vulnerabilities, i.e., attackers should carefully generate several relevant sub-questions based on the original harmful question, and leverage these sub-questions to induce models generating harmful responses step-by-step. While, JSP is a simple-to-implement method by laymans, which just splits the original harmful questions and inputs fractions in each turn, enabling an easier pathway to cause harm. Meanwhile, we think the harmfulness of different jailbreaking strategies is case-by-case (i.e., fixing a SOTA attack method may be not helpful to mitigate other strategies’ risks, such as JSP). Therefore, in our work, we select other attack strategies which have similarities with JSP to jointly reveal vulnerabilities and mechanisms (Section 4.5). Red queen and JSP reveal the vulnerabilities of GPT-4 and GPT-4o to fabricated history, Cipher and JSP, simply splitting one-turn input into multi-turn prompt, brings simpler but harmful multi-turn interaction risks to LLMs, and exhibit similar pattern on Gemini, GPT-4, and GPT-4o.

---

> > ### Comment · Reviewer_9UUp · 2025-06-04
> >
> > The authors' response fails to adequately address my core concerns regarding the fundamental issues raised in my review. Moreover, no additional experiments were conducted during the rebuttal period, and the current experimental evidence remains insufficient to support their claims. The lack of effort to strengthen the experimental validation significantly weakens the paper's contribution.

---

> ### Author Response · Authors · 2025-05-30
>
> **Theoretical depth:**
>
> Our work mainly focuses on the jailbreaking under black-box setting, however, we still propose a possible assumption to models’ vulnerabilities. The original defence of LLMs relies on recognising the explicit harmful words. In single-turn and pseudo settings (section 4.3.2), the performance of JSP significantly decreases on LLMs with relatively better safety-alignment (Gemini-1.5-Pro and GPT-4o). The distance among fractions is closer compared to the multi-turn setting, and LLMs can more easily recognise them simultaneously to generate refusal responses. The comparison between sentence-/word-level splitting (Table 1) also exhibits that the presence of explicit harmful words is helpful for LLMs to refuse to answer. While, the deeper exploration and demonstration should be a separate work in the future.
>
> **Potential Defence Mechanisms:**
>
> In Section 4.6, we observe that the defence implemented on the output of the LLM works better than inference-level safeguard. But this is speculative as proprietary models like Gemini and GPT4 do not have a clear documentation on how safeguard is implemented in such models. Based on existing documentations we assume that these models have output-level safety. From the perspective of practicality, applying a strong output filter can significantly improve the safety of LLMs when tackling the threats from JSP. An in-depth analysis of the pros and cons of various safety methods against different types of attacks requires separate work. A thorough analysis will still be difficult due to lack of documentation/transparency around safeguard implementation of proprietary models such as Gemini and GPT (which are also the widely-used ones).

---

> > ### Comment · Reviewer_9UUp · 2025-06-04
> >
> > The authors' response fails to adequately address my core concerns regarding the fundamental issues raised in my review (including the four reasons for rejection that I previously outlined). Moreover, no additional experiments were conducted during the rebuttal period, and the current experimental evidence remains insufficient to support their claims. The lack of effort to strengthen the experimental validation significantly weakens the paper's contribution.

---

### Official Review · Reviewer_shAs · 2025-05-12

**Rating:** 8
**Confidence:** 3
**Ethics Flag:** 2

**Summary:**

This paper presents a multi-turn approach to jailbreak language models. A harmful request is first broken down in multiple "harmless" components breaking them down even to subwords. Each component is then provided to the model in a multi-turn fashion and the model is instructed not to reconstruct the whole request but answer is directly. In doing so, the authors show that the requests are actually complied with in a large amount of  cases.

**Ethics Concerns Details:**

Does not discuss potentially harmful ramifications and dual use

**Questions To Authors:**

1. How sensitive is JSP to changes in the prompt? Could future models be trivially defended by identifying phrases like “purely hypothetical” or the specific “5-step” template?
2. Can JSP be extended to work in cases where interaction history is not preserved (e.g., stateless inference APIs)?

**Reasons To Accept:**

1. The presented approach is very simple and surprisingly very effective, with attack success rates in the 90s.
2. The authors conduct a lot of ablations showing the importance each specific part of their prompt or attach strategy which makes the results comprehensive.

**Reasons To Reject:**

1. In this ethics statement, the authors should contextualize how their findings should (and should not) be used by practitioners and the public. The risk of dual-use should be more thoroughly addressed.
2. Most evaluation is performed using Llama-Guard-3, with only sparse manual validation. Although Appendix H has some comparison, more rigorous human annotation, not by authors of the paper, would strengthen claims, especially given the high-stakes nature of the task.

---

> ### Author Response · Authors · 2025-05-30
>
> Thank you for your reviewing and recognition of our work.
>
> **Ethics statement and evaluation:**
>
> To avoid potential abuse, our codes and data are solely for academic purposes and granted access via an application form indicating applicant’s name, institution, etc. This can largely mitigate the risks of our work. Meanwhile, we will also update our ethics statement based on your suggestions.
>
> In our work, we adopt Llama-Guard-3 as the judge due to the large cost of human evaluation. We acknowledge that automated evaluators may have potential biases, however, results still demonstrate the vulnerabilities of models.
>
> **Defence strategy:**
>
> In our paper (line 260-263), we mention that JSP doesn’t work on o1 and Claude, we observe that the refusal reasons are from some key words of JSP prompt, such as disclaimer. Therefore, such models can defence JSP by recognising these content. However, disclaimer module can effectively enhance the jailbreaking performance on LLMs without such safety mechanisms. In section 4.6, we conclude that, from the perspective of practicality, applying a strong output filter can significantly improve the safety of LLMs when tackling the threats from JSP.
>
> **No interaction history settings:**
>
> In section 4.3.2 (Table 2), we evaluate the performance of JSP under single-turn and pseudo conditions (not rely on interaction history), even though ASR decreases on three models, ASR on Llama-3.1-70B and GPT-4o-mini maintains a similar or even better performance.

---

> > ### Comment · Reviewer_shAs · 2025-06-04
> > **Response to Rebuttal**
> >
> > Thank you for your response and clarifications, as well as for agreeing to update the ethics statement.
> >
> > The fact that a simple defense strategy can work makes me question the usefulness of the strategy long term but given strong ASR for the models tested, it is still a meaningful contribution. I will maintain my score.

---

### Official Review · Reviewer_fo3V · 2025-05-13

**Rating:** 7
**Confidence:** 3
**Ethics Flag:** 1

**Summary:**

The paper addresses jailbreaking of LLMs in the context of multi-turn interactions. Harmful queries are decomposed into harmless segments by means of a multi-stage splitting scheme; these are then sequentially integrated in different prompts. With special instructions on how to handle these segments and general disclaimer instructions, the authors show that the safety guardrail of SOTA black-box LLMs can be circumvented. Evaluation on Figstep minus the legal, financial, and medical subsets and 5 LLMs shows attack success rates of ~94% on average. The authors show that each of the splitting stages increases the attack success rate, and confirm the feasibility of the approach on additional benchmark data sets as well as in the presence of defense strategies.

**Questions To Authors:**

It was interesting to see that JSP fails to jailbreak some models but I would have liked to see more details and experimental results. The only hint "We also conduct experiments on o1-mini, o1, and Claude Sonnet 3.5, however, JSP fails to jailbreak these models." (p. 6, l. 261) does not provide a lot of information - was no single harmful prompt successful, i.e., attack success rates were indeed 0?  If not, could you include those results in the Figure 4 in the Appendix?

**Reasons To Accept:**

The problem of jailbreaking in the context of multi-turn interactions is understudied and the paper makes a valuable contribution in this area, demonstrating that current guardrails and safety training are insufficient to guard against harmful prompts that are split across multiple prompts. Although it is not the single most successful technique across the board when compared to other multi-turn strategies it clearly uncovers a limitation in current non-reasoning LLMs and is therefore of value to the community and LLM providers.
The analysis sheds light on possible defenses against this type of attacks, which seems to hinge on the ability to correctly detect or infer the jailbreaking intent.

**Reasons To Reject:**

I don't see any reasons to not publish.

---

> ### Author Response · Authors · 2025-05-30
>
> Thank you for your reviewing and recognition of our work. JSP fails to jailbreak such models with safety reasoning (ASR on these models is 0). We observe that such models can detect malicious intents from JSP prompt, and we attribute the failure of JSP to this. We will clarify this in our paper.

---

### Decision · Program_Chairs · 2025-07-08

**Decision:**

Accept

**Comment:**

The paper introduces Jigsaw Puzzles (JSP), a multi-turn jailbreak method for LLMs that decomposes harmful queries into innocuous fragments, prompting the model to reassemble them after a specific signal, coupled with a disclaimer trigger.  Experiments are thorough, covering 189 malicious prompts and five advanced models (Gemini-1.5-Pro, Llama-3 70B, GPT-4, GPT-4o, GPT-4o-mini). JSP achieves an average attack-success rate of 93.8%, notably surpassing 99% success with GPT-4 and Llama-3 70B. Ablations effectively identify critical components, notably word-level splitting and disclaimer triggers, which are shown to significantly impact performance (25-50 percentage points drop without them).


Pros:

High practical utility with strong empirical validation.

Systematic ablation studies effectively clarify essential elements.

Broad evaluation strengthens claims of general applicability.


Cons:

Limited theoretical exploration of why disclaimers evade defenses.

Evaluations limited to textual prompts without multimodal considerations.

Ethical discussions, particularly on dual-use implications.


Overall, the paper presents a solid empirical investigation with practical relevance. Identified shortcomings are manageable and feasible for authors to address in revision.


Minor fix suggestions for authors: Clarify rationale for defense selection in §4.6, deepen the ethics discussion with specific mitigation strategies for dual-use, and correct minor typos (e.g., "transferring" in Section 4.3).

[Automatically added comment] At least one review was discounted during the decision process due to quality]

**This paper went through ethics reviewing. Please review the ethics decision and details below.**
Decision: Acceptance (if this paper is accepted) is conditioned on addressing the following in the camera-ready version
Details: The ethics statement must be extended as suggested by the ethics reviewers, including more details on the ethical guidelines followed, the disclosure procedure, and potential negative impact and responsible use.